# Systemic Inflammatory Response and Outcomes in Community-Acquired Pneumonia Patients Categorized According to the Smoking Habit or Presence of Chronic Obstructive Pulmonary Disease

**DOI:** 10.3390/jcm9092884

**Published:** 2020-09-07

**Authors:** Ernesto Crisafulli, Catia Cillóniz, Adamantia Liapikou, Marcello Ferrari, Fabiana Busti, Domenico Girelli, Antoni Torres

**Affiliations:** 1Department of Medicine, Respiratory Medicine Unit, University of Verona and Azienda Ospedaliera Universitaria Integrata of Verona, 37134 Verona, Italy; ernesto.crisafulli@univr.it (E.C.); marcello.ferrari@univr.it (M.F.); 2Department of Medicine, Section of Internal Medicine, University of Verona and Azienda Ospedaliera Universitaria Integrata of Verona, 37134 Verona, Italy; fabiana.busti@gmail.com (F.B.); domenico.girelli@univr.it (D.G.); 3Pneumology Department, Clinic Institute of Thorax (ICT), Hospital Clinic of Barcelona—Institut d’Investigacions Biomèdiques August Pi i Sunyer (IDIBAPS)—University of Barcelona—Ciber de Enfermedades Respiratorias (CIBERES), 08036 Barcelona, Spain; catiacilloniz@yahoo.com; 46st Respiratory Department, Sotiria Hospital, 11527Athens, Greece; mliapikou@yahoo.com

**Keywords:** community-acquired pneumonia, smoking habit, inflammatory response, outcomes, chronic obstructive pulmonary disease, C-reactive protein

## Abstract

The systemic inflammatory response (SIR) may help to predict clinical progression, treatment failure, and prognosis in community-acquired pneumonia (CAP). Exposure to tobacco smoke may affect the SIR; the role of smoking in CAP has not been consolidated. We evaluated the SIR and outcomes of hospitalized CAP patients stratified by smoking habits and the presence of COPD. This retrospective analysis was conducted at the Hospital Clinic of Barcelona. Baseline, clinical, microbiological, and laboratory variables were collected at admission, using C-reactive protein (CRP) levels as a marker of SIR. The study outcomes were pleural complications, hospital stay, non-invasive and invasive mechanical ventilation (IMV), and intensive care unit (ICU) admission. We also considered the in-hospital and 30-day mortality. Data were grouped by smoking habit (non-, former-, and current-smokers) and the presence of COPD. Current smokers were younger, had fewer comorbidities, and fewer previous pneumonia episodes. CRP levels were higher in current smokers than in other groups. Current smokers had a higher risk of pleural complications independent of CRP levels, the presence of pleuritic pain, and a higher platelet count. Current smokers more often required IMV and ICU admission. Current smokers have a greater inflammatory response and are at increased risk of pleural complications.

## 1. Introduction

Community-acquired pneumonia (CAP), an infection of the pulmonary parenchyma, causes significant mortality and morbidity worldwide [1]. Evaluating the systemic inflammatory response (SIR) in CAP could help to identify the etiological cause [2] and to assess the clinical course [3], including treatment failure [4] and prognosis [5]. The C-reactive protein (CRP), a non-specific acute phase protein produced by the liver in response to interleukin-6 (IL-6) stimulation [6] is considered a valuable serum biomarker for SIR in CAP [7,8]. Patients with CAP and associated chronic obstructive pulmonary disease (COPD) have a lower inflammatory response than their peers without COPD, which is only partially related to corticosteroid treatments [7]. Mechanisms that involve alveolar macrophages and/or phagocytosis may occur in a specific early inflammatory response to infection [9,10].

Several baseline patient characteristics may influence the outcomes of CAP [11]. Although the presence of COPD is a risk factor for CAP [12], being associated with a worse clinical presentation during hospitalization, the CAP-related mortality has been reported to be similar in patients with and without COPD [13,14]. However, this latest finding is controversial [15,16]. Exposure to tobacco smoke is also considered to predispose to the development of a CAP [12]. However, data concerning the impact of smoke on CAP outcomes are inconsistent [17,18,19] due to the association with many tobacco-related host factors in smokers, such as cardiovascular disease [20]. In pneumococcal CAP, an increased mortality risk has been reported in current smokers as compared to former smokers and non-smokers, but the distinct influence of COPD has been not considered [21]. Data about the relationship between the SIR and smoking status are also lacking.

Tobacco exposure appears to increase susceptibility to CAP through several mechanisms [22] that promote respiratory infection by suppressing normal defense functions [23]. Central to this is the increase in cellular oxidative stress, which seems to induce different responses to pathogens by immune cells, including alveolar macrophages and peripheral blood mononuclear cells [24]. In an experimental model, exposure to cigarette smoke induced peribronchial and perivascular lymphocytic aggregates and the parenchymal accumulation of macrophages, thereby upregulating alveolar macrophages inflammation [25]. Moreover, smoke exposure modifies bronchoalveolar lavage cells epigenetically, resulting in a continuous active demethylation and increased inflammatory processes in the lungs [26]. Cigarette smoke affects the innate and adaptive immune responses, increasing numerous pro-inflammatory cytokines (e.g., TNF-α, IL-1, IL-6, and IL-8) [27].

In this study, we evaluated the inflammatory response and outcomes in patients hospitalized with CAP stratified by smoking habits and the presence of COPD.

## 2. Materials and Methods

### 2.1. Design, Patients Enrolled and Definitions

This was a retrospective analysis of two sets of data [7,13] that were collected prospectively from the Hospital Clinic of Barcelona (Spain) between 2004 and 2008. All consecutive adult patients admitted to hospital with a diagnosis of CAP were considered for inclusion, excluding those with nosocomial pneumonia. We also excluded patients with severe immunosuppression, such as those with hematologic malignancy or neutropenia after chemotherapy, bone marrow transplantation, solid-organ transplantation, HIV infection, or using immunosuppressive drugs. The Hospital’s Ethics Committee approved the protocols of each study (2004/1855 [7]–2007/3543 [13]), which were both conducted according to the Good Clinical Practice recommendations and the requirements of the Declaration of Helsinki.

Pneumonia was defined as any new radiographic evidence of pulmonary infiltrate on admission that accompanied signs or symptoms compatible with a lower respiratory tract infection. The diagnosis and severity of COPD were assessed according to the Global Initiative for Chronic Obstructive Lung Disease [28]. Spirometry, confirming the presence of airflow obstruction in patients with COPD, was required to have been performed before admission for CAP. The absence of COPD in former or current smokers was confirmed by clinical history. Smoking was defined as a habit of at least 10 pack-years.

### 2.2. Measurements at Admission

We recorded demographic variables, the prevalence of reported comorbidities (e.g., heart, liver, renal, and neurological disease, plus diabetes mellitus and malignancy), the patient’s usual medications (e.g., use of antibiotics in the last month, inhaled corticosteroids (ICS), oral corticosteroids (OCS), or H_2_ antagonists), and previous anti-pneumococcal and influenza vaccinations. Alcohol abuse was considered when current intake was >80 g/day in men or >60 g/day in women [29].

CAP severity was assessed by the Pneumonia Severity Index (PSI) [30], and SIR was evaluated by measuring CRP levels using a commercially available immunoturbidimetric method (Bayer AG) with a lower detection limit of 0.1 mg/dL. Clinical symptoms, such as chills, cough, pleuritic pain, or confusion were recorded, as were clinical variables, such as body temperature, respiratory rate ≥30 breaths/minute, systolic blood pressure (SBP) ≤90 mmHg, and heart rate ≥120 beats/minute. The results of blood gas analyses (pH, partial arterial carbon dioxide pressure (PaCO_2_), the ratio of partial arterial oxygen pressure to the fraction of inspired oxygen (PaO_2_/FiO_2_)), blood testing (leukocytes, neutrophils, hematocrit, platelets, glucose, creatinine, sodium, and potassium), and pulmonary complications (e.g., multilobar involvement of ≥2 lobes, pulmonary atelectasis, parapneumonic pleural effusion and pulmonary empyema) were also recorded.

Microbiological samples were collected according to a standard protocol. This was conducted as follows: (a) two blood cultures; (b) urine for antigen detection of *Streptococcus pneumoniae* (Binax NOW *S. pneumoniae* Urinary Antigen Test, Emergo Europe, The Netherlands) and Legionella pneumophila serogroup 1 (Binax NOW *L. pneumophila* Urinary Antigen Test; Trinity Biotech plc); (c) sputum specimens; (d) nasopharyngeal swabs for respiratory virus detection; (e) pleural fluid by thoracocentesis; (f) paired serology on admission and in the third or sixth week to detect seroconversion for *Chlamydophila pneumoniae* and *L. pneumophila*, *Coxiella burnetii*, *Chlamydia psittaci*, *Mycoplasma pneumoniae*, and respiratory viruses (e.g., influenza viruses A and B; parainfluenza viruses 1, 2, and 3; respiratory syncytial virus; adenovirus). High serum titers of immunoglobulin M antibodies during the acute phase were accepted for the diagnosis of atypical microorganisms, such as *C. pneumoniae* (≥1:32), *C. burnetii* (≥1:80), and *M. pneumoniae* (any positive titer).

### 2.3. Outcomes

The primary outcome was the development, at admission or in the first days of hospitalization, of a pleural complication (parapneumonic pleural effusion or empyema requiring chest drainage or surgical treatment). Other outcomes concerning the clinical progression of CAP were the length of hospitalization, the use of non-invasive or invasive mechanical ventilation (NIMV and IMV, respectively), and the need for admission to intensive care. Finally, the mortality rates in-hospital and within 30 days of admission were also considered.

### 2.4. Statistical Analysis

The data had a non-normal distribution, and as such, categorical variables are reported as numbers and percentages while continuous variables are reported as medians with interquartile ranges (IQR). Categorical variables were compared by the *χ*^2^ test or the Freeman–Halton extension [31] of the Fisher exact test, while continuous variables were assessed by the non-parametric Kruskal–Wallis *H* or Mann–Whitney *U* tests, as appropriate. All analyses were performed using IBM SPSS, version 25.0 (IBM Corp., Armonk, NY, USA), with *p*-values of <0.05 considered statistically significant unless otherwise stated.

A multivariate regression analysis was used to identify predictors of pleural complications (the dependent variables). Several variables were tested in the univariate model. These included age ≥65 years, gender, smoking habit (i.e., non-smokers, former smokers, current smokers), COPD, comorbidities, PSI classes IV and V, alcohol use, the forced expiratory volume in the first second (FEV_1_, % predicted), pneumonia in the previous year, antibiotic use during the previous month, ICS, OCS, H_2_ antagonists, anti-pneumococcal and influenza vaccination, CRP (≥150 mg/L, indicating patients with a high inflammatory response [32]), temperature (≥39 °C), symptoms (i.e., chills, cough, pleuritic pain, confusion), respiratory rate (≥30 bpm), SBP (≥90 mmHg), heart rate (≥120 bpm), multilobar involvement (≥2 lobes), presence of pulmonary signs (i.e., atelectasis, parapneumonic pleural effusion, or empyema), pH, PaCO_2_, PaO_2_/FiO_2_ (≤200), blood count (i.e., leucocytes, neutrophils, hematocrit, platelets), and blood biochemistry (i.e., glucose, creatinine, sodium, and potassium). Microbiological positivity for *S. pneumoniae* and *L. pneumophila* was also included. Variables that showed an association with a *p*-value < 0.1 were included in the multivariate Cox proportional hazard regression stepwise model, and variables that correlated strongly with each other (r > |±0.3|) were excluded from the multivariate analyses. Unadjusted and adjusted odds ratios (OR) and their 95% confidence intervals (95%CIs) were then calculated. Finally, calibration was assessed with the Hosmer–Lemeshow goodness-of-fit test.

## 3. Results

### 3.1. General, Clinical, and Microbiological Characteristics

Our study sample comprised 1501 patients with CAP, subdivided into four groups: non-smoker (*n* = 626, 42%), former smoker (*n* = 351, 23%), current smoker (*n* = 273, 18%), and COPD (*n* = 251, 17%) (Table 1). Related to the smoking habit COPD patients were former smoker (*n* = 176, 70%), current smoker (*n* = 59, 24%) and non-smoker (*n* = 16, 6%). In comparison to the other groups, current smokers were younger, had greater alcohol use, lower PSI classes, diabetes, medication use (ICS and H_2_ antagonists), and vaccinations (both pneumococcal and influenza), while the COPD group included more males and was characterized by more pneumonia in the previous year, and increased use of ICS and OCS.

Concerning the clinical variables (Table 2), current smokers tended to have more pleuritic pain, tachycardia, pulmonary atelectasis, and empyema on admission. The COPD group less often had multilobar involvement and pleural effusions, but more often presented with tachypnea, blood gas alterations (especially pH and PaCO_2_), and higher hematocrit values. Finally, the microbiology varied in different groups (Table 3). Microbiology was more often positive in current smokers, and *L. pneumophila* was more prevalent among current smokers compared to the COPD cohort, whereas the opposite was true of *P. aeruginosa*. 

### 3.2. Systemic Inflammatory Response

CRP levels were different among study groups (Figure 1). At admission, current smokers had higher CRP levels (median 226.2 mg/L; IQR 209.5 mg/L) than the non-smoker (median 176 mg/L; IQR 194.2 mg/L), former smoker (median 181 mg/L; IQR 194.1 mg/L), or COPD (median 148.9 mg/L; IQR 181 mg/L) cohorts. The difference was also statistically significant between the former smoker and COPD cohorts. Sub-analysis of CRP levels, measured according to the variables that typically affect the inflammatory response [7] (Table 4) showed a mediating effect of several variables: in current smokers, significantly higher CRP values were only present in patients aged >65 years, who had no heart failure, and who did not use ICS and OCS.

### 3.3. Pleural Complications

In total, 198 patients with CAP (13%) developed pleural complications. In comparison to the non-smoker and former smoker groups, the prevalence of pleural complications was significantly higher in the current smokers (21%) and significantly lower in the COPD (8%) (Table 5). The multivariate regression model (Table 6) indicated that current smoking (OR 2.51; 95% CI 1.36 to 4.64), CRP ≥150 mg/L (OR 2.46; 95% CI 1.25 to 4.81), the presence of pleuritic pain (OR 4.88; 95% CI 2.72 to 8.75), and a higher platelet count (OR 1.003; 95% CI 1.001 to 1.006) were associated with a pleural complication. Of note, an increment of +1 10^3^/L in platelet count evaluated as continuous variable has a very small OR but may be relevant for larger changes. These four predictors (current smoking, CRP, pleuritic pain and platelet count) remained after adjusting the model for demographic variables, PSI, and extent of infection.

### 3.4. Other Outcomes

Concerning the other outcomes, NIMV use was lower in non-smokers compared with former smokers and COPD, but the use of IMV was highest in current smokers (compared with all other cohorts). Although the rate of intensive care admission was higher in current smokers, this was only in comparison to the non-smoker and COPD cohorts. We found no differences in length of hospital stay, the in-hospital mortality rate, or the 30-day mortality rate.

## 4. Discussion

In this observational research, we focused on the association between the smoking habits and the outcomes of CAP among hospitalized patients. Our results indicate that current smokers have an increased SIR at admission, as indicated by their CRP levels, and that this is modulated by age, the presence of chronic heart failure, and by the prior use of ICS or OCS therapy. Moreover, current smokers have an increased chance of having a pleural complication during hospitalization, independent of the higher SIR or the presence of pleuritic pain.

Tobacco exposure increases the susceptibility to CAP [22], promoting respiratory infection [23]. Cigarette smoke affects the innate and adaptive immune responses with an increase in numerous pro-inflammatory cytokines [27]. In our research, we could document an increased inflammatory response in the current smoker group based on higher CRP levels.

In addition, we showed that being a current smoker increased the risk of pleural complication. Again, there seems to be a specific susceptibility among current smokers, with smoke having a local impact that is not fully controlled by the inflammatory response. The hypothesis that smoke has a direct effect on pleural complications may be derived from our model adjusted for CAP severity (PSI) and infection extent (multilobar involvement). In current smokers, the lower prevalence of neurological disease, ICS use [33], and influenza vaccination, together with the younger age, all of which are protective against pleural complications, may have played a role. It was also interesting to note that, despite their higher prevalence of pleural complications, current smokers had similar lengths of hospital stay to the other patients, confirming that these patients still achieve a good clinical response.

The impact of smoking on the outcomes of CAP have been evaluated but not consolidated [17,18,19]. Although large longitudinal studies indicate that there is an increased risk of pneumonia-associated death among smokers compared with non-smokers, this has not been confirmed by cross-sectional studies [34]. Surprisingly, smoking seems to have a counterintuitive protective role on long-term mortality [35]. Similar to the previous comparison between patients with and without COPD [13,14], we did not find differences in prognosis by smoking habit or the presence of COPD. Current smokers did have a higher likelihood of needing IMV or intensive care admission, but this was probably related not only to the severity of clinical condition (i.e., tachycardia, multilobar involvement, and pleural complications were more prevalent) but also to their younger age, necessitating a more cautious approach [36]. Although smoking has been reported to increase the 30-day mortality rate [21], COPD was not considered as a distinct group, having a different inflammatory response [7,8,13], possibly mediated by different alveolar macrophages activation [9,10] that could influence the outcomes of CAP [3,4,5].

Finally, concerning the microbiological causes of CAP, smoking was shown as an independent risk factor for Legionnaires’ disease [37], again due to the associated impairment of defensive mechanisms of respiratory system [22,38]. For the same reason, patients with COPD, especially those with advanced disease, were noted to be more susceptible to *P. aeruginosa* [39].

The major strengths of this study were the inclusion of a large cohort of patients, and the use of spirometry to objectively confirm all diagnoses of COPD before inclusion. However, relatively few spirometric exams were performed for patients without COPD, so we cannot exclude airflow obstruction despite patients reporting no symptoms compatible with COPD in their clinical history (e.g., dyspnea). The single-center design and use of only CRP as the inflammatory marker also represent important limitations. In this context, our data sets have been collected in an anterior period to the spreading of the multiplex PCR assays, such as the microbial investigation, suboptimal according to the current state of the art. Therefore, the role of inflammation in smokers with CAP needs to be confirmed by several mediators in future prospective research. Similarly, prospective data about prediction tools, such as the Sequential Organ Failure Assessment (SOFA) score, could better define the severity of CAP [40] and any possible correlations with smoking habits.

In conclusion, our study demonstrates that current smokers, in comparison to non-smokers, former smokers, and COPD, have a higher SIR to CAP. Current smokers represent a category of CAP patients at risk for pleural complications. Preventive strategies should target these patients.

## Figures and Tables

**Figure 1 jcm-09-02884-f001:**
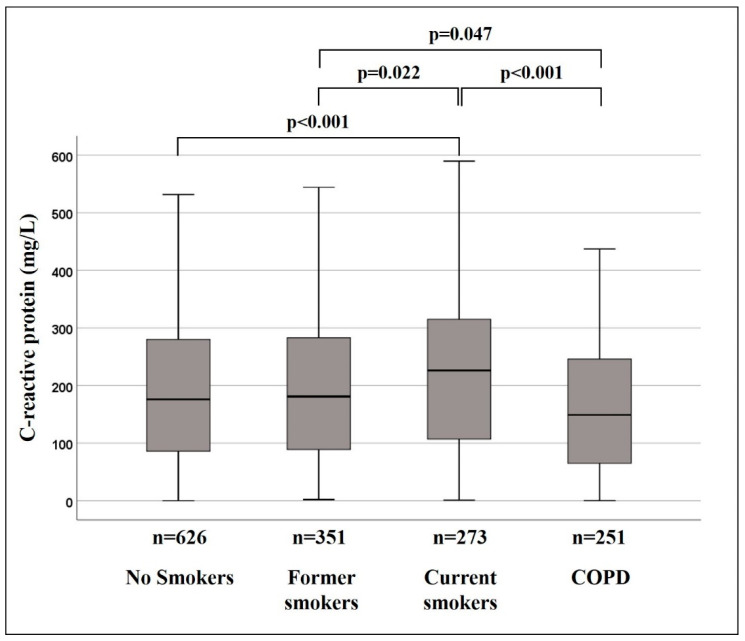
C-reactive protein levels of study cohorts.

**Table 1 jcm-09-02884-t001:** Baseline characteristics of patients with CAP by smoking habit and COPD diagnosis.

Variables	Non-Smokers	Former Smokers	Current Smokers	COPD	*p* Value
Number of patients	626	351	273	251	
Age, years	77 [21]	75 [16]	53 [24] **^,§§^	75 [11] ^##^	<0.001
Male, %	30	87 **	72 **^,§§^	92 **^,##^	<0.001
PSI, classes IV+V, %	63	70 *	42 **^,§§^	71 *^,##^	<0.001
History of alcohol, %	4	26 **	38 **^,§^	28 **^,#^	<0.001
FEV_1_, % predicted				48 [22]	
Chronic heart failure, %	24	30 *	8 **^,§§^	25 ^##^	<0.001
Chronic liver disease, %	3	3	6	5	0.086
Chronic renal failure, %	5	11 *	4 ^§^	7	0.002
Neurological disease, %	33	23 *	16 **	23 *	<0.001
Diabetes mellitus, %	20	25	13 *^,§§^	23 ^#^	0.003
Malignancy, %	4	9 *	3 ^§^	13 **^,##^	<0.001
Pneumonia during previous year, %	17	22	13 ^§^	42 **^,§§,##^	<0.001
Previous antibiotic in the last month, %	30	31	25	37	0.090
Previous ICS therapy, %	12	20 *	4 *^,§§^	65 **^,§§,##^	<0.001
Previous OCS therapy, %	4	6	3	12 **^,§,##^	<0.001
Previous use of H_2_ antagonists, %	16	16	8 *^,§^	15 ^#^	0.011
Antipneumococcal vaccination					<0.001
No	81	75	93 *^,§§^	70 **^,§,##^	
<5 years, %	3	5	0.5	0.6	
≥5 years, %	16	20	6.5	29.4	
Influenza vaccination					<0.001
No	40	37	71 **^,§§^	26 *^,##^	
<6 months, %	39	44	16	49	
≥6 months, %	21	19	13	25	

Data are shown as percentages or as medians [interquartile range]. Percentages are calculated for non-missing data. * *p* < 0.05 and ** *p* < 0.001 versus non-smokers; ^§^
*p* < 0.05 and ^§§^
*p* < 0.001 versus former smokers; ^#^
*p* < 0.05 and ^##^
*p* < 0.001 versus current smokers. Abbreviations: COPD, chronic obstructive pulmonary disease; FEV_1_, forced expiratory volume in the first second; ICS, inhaled corticosteroids; OCS, oral corticosteroids; PSI, Pneumonia Severity Index.

**Table 2 jcm-09-02884-t002:** Clinical and laboratory variables by smoking habit and COPD diagnosis.

Variables	Non-Smokers	Former Smokers	Current Smokers	COPD	*p* Value
Temperature ≥ 39 °C, %	13	10	17	12	0.093
Chills, %	54	60	59	58	0.225
Cough, %	79	81	83	84	0.364
Pleuritic pain, %	40	40	55 **^,§§^	43 ^#^	<0.001
Confusion, %	28	23	21	20	0.067
Respiratory rate ≥ 30 bpm, %	28	28	30	38 *^,§^	0.044
SBP ≤ 90 mmHg, %	6	5	6	4	0.778
Heart rate ≥ 120 bpm, %	13	14	24 **^,§^	17 ^#^	<0.001
Multilobar involvement (≥2 lobes), %	27	30	36 *	19 *^,§,##^	<0.001
Pulmonary atelectasis, %	2	2	6 *^,§^	3	0.006
Parapneumonic pleural effusion, %	17	15	20	10 * ^#^	0.015
Pulmonary empyema, %	5	3	11 **^,§^	1 ^#^	<0.001
pH	7.46 [0.07]	7.46 [0.06]	7.46 [0.08]	7.44 [0.07] **^,§§,#^	<0.001
PaCO_2_, mmHg	35.3 [8.4]	33.8 [6.7]	32.9 [7.2]	37.8 [11.5] **^,§§,##^	<0.001
PaO_2_/FiO_2_	280.9 [76.2]	276.2 [81.1]	280.9 [90.5]	265.8 [79.9] *	0.023
Leucocytes, 10^3^/L	12.7 [8.6]	12.35 [7.9]	13.5 [8]	13.4 [7.7]	0.142
Neutrophils, %	83 [11]	83 [11]	82 [10]	83 [9.5]	0.932
Hematocrit, %	39 [6]	40 [6] *	41 [7] *	42 [7] **^,§,#^	<0.001
Platelets, 10^3^/L	237 [112]	221 [117]	241 [119]	239 [128]	0.056
Glucose, mg/dL	127 [56]	125 [54]	120 [52]	123 [60]	0.682
Creatinine, mg/dL	1 [0.5]	1.2 [0.7] *	1 [0.5]	1 [0.4]	0.028
Sodium, mEq/L	135 [6]	135 [6]	134 [7.4] *^,§^	136 [5] ^#^	0.001
Potassium, mEq/L	4 [0.9]	4.1 [0.7]	3.9 [0.6] ^§^	4.1 [0.7] *^,##^	<0.001

Data are shown as percentages or medians [interquartile range]. Percentages are calculated for non-missing data. * *p* < 0.05 and ** *p* < 0.001 versus non-smokers; ^§^
*p* < 0.05 and ^§§^
*p* < 0.001 versus former smokers; ^#^
*p* < 0.05 and ^##^
*p* < 0.001 versus current smokers. Abbreviations: COPD, chronic obstructive pulmonary disease; PaCO_2_, partial pressure of arterial carbon dioxide; PaO_2_/FiO_2_, ratio of partial pressure of arterial oxygen to the fraction of inspired oxygen; SBP, systolic blood pressure.

**Table 3 jcm-09-02884-t003:** Microbiological variables by smoking habit and COPD diagnosis.

Variables	Non-Smokers	Former Smokers	Current Smokers	COPD	*p* Value
Patients with etiological diagnosis, *n* (%)	255 (41)	141 (40)	153 (56) **^,§§^	111 (44) ^#^	<0.001
*Streptococcus pneumoniae*, *n* (%)	116 (45.4)	70 (50)	72 (47.1)	55 (49.5)	0.832
*Streptococcus viridans*, *n* (%)	6 (2.3)	3 (2.1)	3 (2)	4 (3.6)	0.837
*Staphylococcus aureus*^a^, *n* (%)	19 (7.4)	8 (5.7)	7 (4.6)	3 (2.7)	0.293
*Staphylococcus* spp., *n* (%)	5 (2)	5 (3.5)	3 (2)	3 (2.7)	0.763
*Haemophilus influenzae*, *n* (%)	9 (3.5)	2 (1.4)	3 (2)	3 (2.7)	0.590
*Haemophilus parainfluenzae*, *n* (%)	1 (0.4)	1 (0.7)	1 (0.6)	0 (0)	0.943
*Moraxella catarrhalis*, *n* (%)	0 (0)	1 (0.7)	1 (0.6)	2 (1.8)	0.554
*Legionella pneumophila*, *n* (%)	14 (5.5)	17 (12.1) *	23 (15) *	4 (3.6) ^§,#^	<0.001
Other atypical pathogens ^b^, *n* (%)	5 (2)	2 (1.4)	1 (0.6)	1 (0.9)	0.698
*Pseudomonas aeruginosa*, *n* (%)	12 (4.7)	3 (2.1)	3 (2)	13 (11.7) *^,§,#^	<0.001
*Klebsiella pneumoniae*, *n* (%)	4 (1.6)	1 (0.7)	0 (0)	0 (0)	0.786
*Moraxella catarrhalis*, *n* (%)	0 (0)	1 (0.7)	1 (0.6)	2 (1.8)	0.554
*Escherichia coli*, *n* (%)	5 (2)	2 (1.4)	2 (1.31)	2 (1.8)	0.956
Respiratory virus ^c^, *n* (%)	25 (9.8)	15 (10.6)	16 (10.5)	7 (6.3)	0.634
Polymicrobial pneumonia, *n* (%)	20 (7.8)	18 (12.8)	20 (13.1)	15 (13.5)	0.220

Data about pathogens are reported as number of patients (percentage) relative to patients with etiological diagnosis in each cohort. ^a^ Including methicillin-sensitive and methicillin-resistant *S. aureus*; ^b^ including *Mycoplasma pneumoniae*, *Coxiella burnetii*, and *Chlamydia*; ^c^ including influenza A, influenza B, parainfluenza virus, respiratory syncytial virus, and adenovirus. * *p* < 0.05 and ** *p* < 0.001 versus non-smokers; ^§^
*p* < 0.05 and ^§§^
*p* < 0.001 versus former smokers; ^#^
*p* < 0.05 versus current smokers. Abbreviations: COPD, chronic obstructive pulmonary disease.

**Table 4 jcm-09-02884-t004:** Inflammatory response by smoking habit and COPD diagnosis for the main variables.

Variables	Categories	Non-Smokers	Former Smokers	Current Smokers	COPD	*p* Value
Age	<65 years	*n* = 169	*n* = 78	*n* = 199	*n* = 43	0.131
	CRP value, mg/L	183.9 [205.5]	187.4 [246.2]	230 [216]	192.7 [246]	
	≥65 years	*n* = 457	*n* = 273	*n* = 74	*n* = 208	0.001
	CRP value, mg/L	173 [193.1]	179 [181.5]	208.4 [190]	135.2 [170.3] ^##^	
Chronic heart failure	No	*n* = 476	*n* = 244	*n* = 250	*n* = 189	<0.001
	CRP value, mg/L	187.1 [194.5]	185.4 [194.1]	224.2 [199.3] *	172 [178] ^§,##^	
	Yes	*n* = 148	*n* = 107	*n* = 21	*n* = 62	0.278
	CRP value, mg/L	131.3 [193.4]	153 [190.8]	237.9 [251.1]	112.2 [193.6]	
Previous ICS therapy	No	*n* = 547	*n* = 279	*n* = 255	*n* = 86	0.001
	CRP value, mg/L	179 [192.7]	181 [195]	226.2 [198.1] *	156.4 [176.7] ^#^	
	Yes	*n* = 73	*n* = 69	*n* = 12	*n* = 162	0.607
	CRP value, mg/L	130 [210.3]	174 [183.6]	189.2 [183.8]	140.7 [190.5]	
Previous oral corticosteroid therapy	No	*n* = 491	*n* = 267	*n* = 205	*n* = 217	<0.001
	CRP value, mg/L	184.2 [196.2]	187.5 [200]	239 [173.8] **^,§^	152 [183] ^§,##^	
	Yes	*n* = 20	*n* = 18	*n* = 6	*n* = 29	0.101
	CRP value, mg/L	104.6 [180]	100 [103.9]	201.7 [170.3]	172.7 [206.3]	

Data are shown as medians [interquartile range]. * *p* < 0.05 and ** *p* < 0.001 versus non-smokers; ^§^
*p* < 0.05 versus former smokers; ^#^
*p* < 0.05 and ^##^
*p* < 0.001 versus current smokers. Abbreviations: COPD, chronic obstructive pulmonary disease; CRP, C-reactive protein; ICS, inhaled corticosteroids.

**Table 5 jcm-09-02884-t005:** Comparison of study outcomes by smoking habit and COPD diagnosis.

Variables	Non-Smokers	Former Smokers	Current Smokers	COPD	*p* Value
Pleural complications, %	13	13	21 *^,§^	8 * ^§,##^	<0.001
Length of hospital stay, days	8 [5]	8 [6]	8 [8]	8 [6]	0.099
NIMV, %	3	7 *	4	8 *	0.033
IMV, %	4	5	10 *^,§^	4 ^#^	0.007
ICU admission, %	7	10	15 **	8 ^#^	0.004
In-hospital mortality, %	6	4	3	3	0.107
30-days mortality, %	7	6	3	4	0.056

Data are shown as percentages or as medians [interquartile range]. Percentages are calculated for non-missing data. * *p* < 0.05 and ** *p* < 0.001 versus non-smokers; ^§^
*p* < 0.05 versus former smokers; ^#^
*p* < 0.05 and ^##^
*p* < 0.001 versus current smokers. Abbreviations: COPD, chronic obstructive pulmonary disease; ICU, intensive care unit; IMV, invasive mechanical ventilation; NIMV, non-invasive mechanical ventilation.

**Table 6 jcm-09-02884-t006:** Univariate and multivariate analysis for predicting the probability of a pleural complication.

Variables	Univariate			Multivariate			Multivariate Adjusted *		
	OR	95% CI	*p* Value	OR	95% CI	*p* Value	OR	95% CI	*p* Value
Non-smokers	1			1			1		
Former smokers	1.009	0.68 to 1.49	0.966	1.42	0.71 to 2.85	0.324	1.44	0.68 to 3.05	0.344
Current smokers	1.75	1.20 to 2.55	0.004	2.51	1.36 to 4.64	0.003	2.52	1.29 to 4.95	0.007
COPD	0.54	0.32 to 0.93	0.026	0.55	0.18 to 1.75	0.315	0.52	0.15 to 1.78	0.299
Age, ≥65 y	0.53	0.39 to 0.72	<0.001						
Neurological disease, yes	0.67	0.43 to 1.03	0.071						
Pneumonia during previous year, yes	0.69	0.46 to 1.03	0.069						
Previous ICS therapy, yes	0.38	0.24 to 0.62	<0.001						
Influenza vaccination, No	1								
<6 months	0.44	0.28 to 0.68	<0.001						
≥6 months	0.67	0.41 to 1.10	0.106						
CRP, ≥150 mg/L	1.50	1.10 to 2.06	0.012	2.46	1.25 to 4.81	0.009	2.58	1.30 to 5.10	0.007
Temperature, ≥39 °C	0.47	0.26 to 0.82	0.008						
Heart rate, ≥120 bpm	1.61	1.11 to 2.35	0.013						
Pleuritic pain, yes	3.71	2.67 to 5.15	<0.001	4.88	2.72 to 8.75	<0.001	5.13	2.78 to 9.46	<0.001
Positivity for *S. pneumoniae*	1.50	1.09 to 2.05	0.012						
Platelets, +1 10^3^/L	1.002	1.000 to 1.003	0.026	1.003	1.001 to 1.006	0.006	1.003	1.001 to 1.006	0.004

* Model adjusted for demographic variables, CAP severity (i.e., PSI), and infection extent (i.e., multilobar involvement). Hosmer–Lemeshow test: *p* = 0.308 in the multivariate model; *p* = 0.510 in the multivariate adjusted models. Abbreviations: CAP, community-acquired pneumonia; CI, confidence interval; COPD, chronic obstructive pulmonary disease; CRP, C-reactive protein; ICS, inhaled corticosteroids; OR, odds ratio; PSI, Pneumonia Severity Index.

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
