# Peer review of "Systemic Inflammatory Response and Outcomes in Community-Acquired Pneumonia Patients Categorized According to the Smoking Habit or Presence of Chronic Obstructive Pulmonary Disease"

_jcm, 2020, doi:10.3390/jcm9092884_

Round 1
Reviewer 1 Report
The authors provide a relevant analysis of retrospective data on systemic inflammatory response in community acquired pneumonia and its association with smoking. The manuscript is generally well written, results are presented appropriately, and interpretation is largely adequate.
There are some minor issues:
The authors have chosen the well established PSI for assessing severity of CAP. Some current literature seems to show superiority of the SOFA score for this purpose. The authors should discuss their choice.
Similarly, the authors chose CRP as a measure of systemic inflammation. What are the arguments for not using PCT, ESR or fibrinogen?
In section "2.2 Pleural Complications", page 10, lines 187 to 190 one may get the impression that current smoking, the presence of pain and a higher platelet count are causing an increased probability of having a pleural complication. While this may actually be the case, the retrospective data only show that there is an association, not a causality. Also, while given correctly in table 6, it may be helpful to explain in words that increment of +1x10^3 in platelet count has a very small OR but that this may become relevant for larger changes.
Similarly, in the beginning of the discussion, page 12, line 212, "... impact of smoking habits on the outcomes..." may be read as analysis of causation, which it probably is not (despite the fact that smoking comes prior to CAP). In line 214, "modulated by" may be more fitting than "mediated by".
To me it appears, that the second paragraph of the discussion, page 12, lines 218 through 227, would be more fitting in the introduction as a motivation to study cigarette smoking in relation to CAP? May be this could then be briefly referred to in the discussion of the authors finding of increased SIR in current smokers?
The section "Acknowledgements" is still containing just the explanatorysentences given by the journal. Is there nobody to acknowledge, patients or any supporting persons or facilities?
In line 264, page 13, "comparison of" should probably be "comparison to"?
Author Response
The authors provide a relevant analysis of retrospective data on systemic inflammatory response in community acquired pneumonia and its association with smoking. The manuscript is generally well written, results are presented appropriately, and interpretation is largely adequate.
We thank the Reviewer for his/her positive comment.
The authors have chosen the well established PSI for assessing severity of CAP. Some current literature seems to show superiority of the SOFA score for this purpose. The authors should discuss their choice.
We thank the Reviewer for his/her comment. SOFA score have been extensively used in emergency department or ICU-area to define better a severe CAP, needing a critical approach. Recent but few evidences report its use as a prognostic tool for non-severe CAP. However, our datasets in our retrospective analysis refers to non-severe CAP, collected some years ago, when the PSI had a unique and very important value; for this reason, we are unable to have in our database data concerning single values of SOFA (for example bilirubin levels or Glasgow Coma Scale score). We have added a sentence about this limitation.
Similarly, the authors chose CRP as a measure of systemic inflammation. What are the arguments for not using PCT, ESR or fibrinogen?
We thank the Reviewer for his/her comment. However, as reported in the previous question, we are unable to have for all patients data concerning PCT, ESR or fibrinogen. This limitation has been reported in the specific section.
In section "2.2 Pleural Complications", page 10, lines 187 to 190 one may get the impression that current smoking, the presence of pain and a higher platelet count are causing an increased probability of having a pleural complication. While this may actually be the case, the retrospective data only show that there is an association, not a causality.
We thank the Reviewer for his/her positive suggestion. In the revised version of the manuscript we have changed the sentence about increased probability of having a pleural complication. As suggested, we have changed it as “association”.
Also, while given correctly in table 6, it may be helpful to explain in words that increment of +1x10^3 in platelet count has a very small OR but that this may become relevant for larger changes.
We thank the Reviewer for his/her positive suggestion. We have added a sentence about this aspect.
Similarly, in the beginning of the discussion, page 12, line 212, "... impact of smoking habits on the outcomes..." may be read as analysis of causation, which it probably is not (despite the fact that smoking comes prior to CAP). In line 214, "modulated by" may be more fitting than "mediated by".
To me it appears, that the second paragraph of the discussion, page 12, lines 218 through 227, would be more fitting in the introduction as a motivation to study cigarette smoking in relation to CAP? May be this could then be briefly referred to in the discussion of the authors finding of increased SIR in current smokers?
We thank the Reviewer for his/her positive comment. We have changed the text accordingly the reviewers’suggestions.
The section "Acknowledgements" is still containing just the explanatory sentences given by the journal. Is there nobody to acknowledge, patients or any supporting persons or facilities?
In line 264, page 13, "comparison of" should probably be "comparison to"?
We thank the Reviewer for his/her comment. We have changed the text, accordingly.
Reviewer 2 Report
Crisafulli et al. did a retrospective analysis of data sets obtained between 2004 and 2008 as a large single center study. The question analysed is about outcome and systemic response in CAP and data were categorized according to smoking (current vs. former) and COPD (yes or no). This is a solid analysis which comes along without surprise ( current smoker do worse), COPD-patients have more Pseudomonas, etc.). Only CRP is used as a marker for systemic inflammatory response.
Author Response
Crisafulli et al. did a retrospective analysis of data sets obtained between 2004 and 2008 as a large single center study. The question analysed is about outcome and systemic response in CAP and data were categorized according to smoking (current vs. former) and COPD (yes or no). This is a solid analysis which comes along without surprise ( current smoker do worse), COPD-patients have more Pseudomonas, etc.). Only CRP is used as a marker for systemic inflammatory response.
We thank the Reviewer for his/her positive comment.
Reviewer 3 Report
Crisafulli et al reported a large cohort of CAP and explored the systemic inflammatory response according to smoking habit +/- COPD. The main finding is that smokers have: higher inflammatory response, increased risk of pleural complication, higher rate of ICU admission.
The strengths are:
- The clarity of the manuscript. The authors have to be congratulated for that.
- The number of patient analyzed (1501 patients). It is a very large cohort, even it is a retrospective analyze of two sets of data.
- The finding that pleural complications was significantly higher in the current smokers (adjusted for demographic variables - among others-)
The limitations are:
- The data sets are old and anterior to the spreading of the multiplex PCR assays. The microbial investigation is suboptimal according to the current state of art (to be acknowledged in the limitations).
- The systemic inflammatory response is only assessed through the CRP. Although CRP is a surrogate for IL-6 in the clinic, there is plenty of other circulating inflammatory pathways (Th17 cytokines, interferons, IL1/IL18, IL8,…). It is questionable whether one can claim that the inflammation is higher with information restricted to CRP (to be discussed more extensively in the limitation section in my opinion).
- There can be some degree of overlap between COPD patients and former/current smokers as spirometry is not available for all patients (not COPD patients). However, it is a classical limitation in this field that is already acknowledged by the authors.
This report is informative and useful for our scientific community.
Author Response
Crisafulli et al reported a large cohort of CAP and explored the systemic inflammatory response according to smoking habit +/- COPD. The main finding is that smokers have: higher inflammatory response, increased risk of pleural complication, higher rate of ICU admission.
The strengths are:
The clarity of the manuscript. The authors have to be congratulated for that.
The number of patient analyzed (1501 patients). It is a very large cohort, even it is a retrospective analyze of two sets of data.
The finding that pleural complications was significantly higher in the current smokers (adjusted for demographic variables - among others-)
We thank the Reviewer for his/her positive comment.
The limitations are:
The data sets are old and anterior to the spreading of the multiplex PCR assays. The microbial investigation is suboptimal according to the current state of art (to be acknowledged in the limitations).
We thank the Reviewer for his/her comment. In the limitation section we have added the suggestions.
The systemic inflammatory response is only assessed through the CRP. Although CRP is a surrogate for IL-6 in the clinic, there is plenty of other circulating inflammatory pathways (Th17 cytokines, interferons, IL1/IL18, IL8,…). It is questionable whether one can claim that the inflammation is higher with information restricted to CRP (to be discussed more extensively in the limitation section in my opinion).
We thank the Reviewer for his/her comment. Our limit is clearly related to the retrospective nature of the study. This aspect has been reported as an important limitation: “…use of only CRP as the inflammatory marker also represent important limitations”. Moreover, we have reported the need for several mediators in further studies: “Therefore, the role of inflammation in smokers with CAP needs to be confirmed by several mediators in future prospective research”.
There can be some degree of overlap between COPD patients and former/current smokers as spirometry is not available for all patients (not COPD patients). However, it is a classical limitation in this field that is already acknowledged by the authors. This report is informative and useful for our scientific community.
We thank the Reviewer for his/her comment. The information required have been reported in the results, first paragraph.